# Gender difference in ASAS HI among patients with ankylosing spondylitis

**Hsin-Hua Chen**[1,2,3,4,5,6,7]*, **Yi-Ming Chen**[1,2,3,4], **Kuo-Lung Lai**[2], **Tsu-Yi Hsieh**[2,8,9], **Wei-Ting Hung**[2,8], **Ching-Tsai Lin**[2], **Chih-Wei Tseng**[2], **Kuo-Tung Tang**[2,3], **Yin-Yi Chou**[2,6], **Yi-Da Wu**[2], **Chin-Yin Huang**[6], **Chia-Wei Hsieh**[2,3], **Wen-Nan Huang**[2,3], **Yi-Hsing Chen**[2,3]*

**1** Department of Medical Research, Taichung Veterans General Hospital, Taichung, Taiwan, **2** Division of Allergy, Immunology, and Rheumatology, Department of Internal Medicine, Taichung Veterans General Hospital, Taichung, Taiwan, **3** School of Medicine, National Yang-Ming University, Taipei, Taiwan, **4** Institute of Biomedical Science and Rong-Hsing Research Center for Translational Medicine, Chung-Hsing University, Taichung, Taiwan, **5** Institute of Public Health and Community Medicine Research Center, National Yang-Ming University, Taipei, Taiwan, **6** Department of Industrial Engineering and Enterprise Information, Tunghai University, Taichung, Taiwan, **7** Institute of Medicine, Chung Shan Medical University, Taichung, Taiwan, **8** Department of Medical Education, Taichung Veterans General Hospital, Taichung, Taiwan, **9** PhD Program of Business, College of Business, Feng Chia University, Taichung, Taiwan

* shc5555@hotmail.com (HHC); ysanne@vghtc.gov.tw (YHC)

## Abstract

### Objective

To assess the associations of the Assessment of Spondyloarthritis International Society Health Index (ASAS HI) with gender and other factors in patients with ankylosing spondylitis (AS).

### Methods

From November 2017 to October 2018, we measured the Ankylosing Spondylitis Disease Activity Score (ASDAS), the Bath Ankylosing Spondylitis Disease Activity Index (BASDAI), the Bath Ankylosing Spondylitis Functional Index (BASFI), the modified Stoke Ankylosing Spondylitis Spinal Score (mSASSS) and the ASAS HI score for AS patients at the Taichung Veterans General Hospital. After adjusting for disease activity (ASDAS-erythrocyte sedimentation rate [ESR], ASDAS- C-reactive protein [CRP], BASDAI+ESR or BASDAI+CRP), mSASSS and other potential confounders including medications, comorbidities, and laboratory data, any associations between gender and the sum score of ASDAS HI were assessed using multiple linear regression analysis, as well as any associations between gender and an ASAS HI score >5 using multivariable logistic regression analysis.

### Results

A total of 307 AS patients (62 [20.2%] females, mean age 46.4 years [S.D. 13.3], mean symptom duration 20.6 years [S.D. 12.1]) were included. Multiple linear regression analysis showed that the male gender was significantly associated with a lower ASAS HI ($B$ = -1. 91, 95% confidence interval [CI], −2.82–−1.00, p <0.001). Multivariable logistic regression analysis revealed that males also had a lower risk of achieving scores of ASAS HI > 5 than

**Data Availability Statement:** All relevant data are within the manuscript and its Supporting Information files.

**Funding:** The authors received no financial support for this work.

 

**Competing interests:** The authors have declared that no competing interests exist.

females (odds ratio = 0.15, 95% CI, 0.07–0.36, p <0.001). Disease activity measures, including ASDAS-ESR, ASDAS-CRP and BASDAI, had positive correlations with ASAS HI.

## Conclusion

This single-center, cross-sectional study revealed that a higher ASAS HI score was significantly associated with female gender and higher disease activity measures.

## Introduction

Ankylosing spondylitis (AS) is a common chronic inflammatory rheumatic disease, which mainly affecting the axial skeleton, but can also affect peripheral joint and entheses [1]. It has a prevalence rate between 0.11% and 0.38% in Taiwan [2, 3]. In recent decades, AS has been thought to be a variety of spondyloarthritis (SpA), which is associated with other extra-articular manifestations (EAMs) including uveitis, psoriasis, and inflammatory bowel disease. AS usually results in impairment of function and diminished health-related quality of life (QoL) [4–9]. AS had long been considered as a disease that occurred mostly in men, with a male to female ratio of around 9:1 [10, 11]. However, recent studies have found the ratio of affected men to women to be about 2–3:1 [3, 12, 13]. Understanding whether the clinical characteristics of AS and treatment outcomes differ between males and females can help rheumatologists to improve quality of care. Various previous studies have investigated gender-attributable differences in AS patients. Compared with males, female AS patients were found to have a higher average age at onset of symptoms [14, 15], a longer delay before AS diagnosis [16], a higher prevalence of fibromyalgia [17], and a higher prevalence of widespread pain [18], which was related to delay in diagnosis [19]. Regarding clinical manifestations, conflicting results were found with respect to gender-related differences in the prevalence of uveitis [15, 19, 20] and peripheral arthritis [15, 19, 21]. With respect to AS-related clinical outcomes, prior studies showed that females had worse outcomes with regards to Bath AS Disease Activity Index (BASDAI), of which total back pain and longer morning stiffness duration showing the largest difference [19, 22–24], function [19, 25], QoL [24–27], and overall well-being in daily life indicated by Bath Ankylosing Spondylitis Global scores. Prior studies revealed an association between the delay in AS diagnosis and worse function [28, 29], which might explain, at least in part, a worse function in female patients with AS compared with male patients with AS. However, men had worse radiographic spinal progression compared to women [15, 16, 21, 27, 30–32].

Recently, expert members of the Assessment of Spondyloarthritis International Society (ASAS) have developed a disease-specific questionnaire named the ASAS Health Index (ASAS HI) based on the Comprehensive International Classification of Functioning, Disability and Health (ICF) Core Set, aimed to better quantify health in SpA [33]. The ASAS HI is a linear composite measure that is comprised of seventeen components, with options to respond "I agree" or "I do not agree". The total aggregate score achievable on the ASAS HI ranges from 0 to 17, with a higher score signifying a worse health status. Although the ASAS HI was originally developed in English, it has been successfully translated in to fifteen languages including Chinese [34]. ASAS HI has been validated to assess health and function in patients with SpA [35]. However, whether the sum scores of ASAS HI differs between male and female AS patients remains unknown. Starting in November 2016, we regularly assessed clinical characteristics, comorbidities and clinical outcomes including disease activity (BASDAI, AS Disease

Activity Score [ASDAS] with erythrocyte sedimentation rate [ESR], and ASDAS with C-reactive protein [CRP]), physical function (Bath AS Functional Index [BASFI]), radiographic damage (modified Stoke AS Spinal Score [mSASSS]) [36] and the ASAS HI for AS patients using an electronic medical record management system developed at the Taichung Veterans General Hospital. The aim of this study was to investigate the influence of gender on the ASAS HI using a Taiwanese AS cohort.

## Methods

### Ethics approval and consent to participate

The Institutional Review Board (IRB) I of Taichung Veterans General Hospital (TCVGH) (IRB TCVGH number: CE18321A) permitted this study. Informed consent was waived because tracked personal information had been anonymized before data analyses.

### Study design

The is a retrospective, single center, cross-sectional study.

### Data collection

In November 2016, we developed an electronic medical record management system at the TCVGH to asssist rheumatologists to comprehensively assess disease characteristics, personal history, family history, comorbidities, and clinical outcomes for AS patients in daily clinical practice. Based on the 1984 modified New York criteria for AS, AS patients were registered after confirmation of AS diagnosis by rheumatologists at the TCVGH (TCVGH-AS cohort) [37]. After the patients were registered, a trained nurse assisted rheumatologists in collecting information including clinical characteristics, age at onset of symptoms, smoking status, previous tuberculosis history, family histories, comorbidities (hypertension, diabetes mellitus, hyperlipidemia, hepatitis B, hepatitis C, gout, coronary artery disease, stroke, periodontal disease and osteoporosis) and extra-spinal manifestation (i.e., the presence or absence of EAMs [uveitis, psoriasis and inflammatory bowel disease], peripheral arthritis, enthesitis, and dactylitis) at presentation. The rheumatologist-in-charge then confirmed the clinical characteristics. The trained nurse also helped AS patients to complete the questionnaire required for the assessments of BASDAI, ASDAS, BASFI, and ASAS HI. Data for BASDAI, ASDAS-ESR, ASDAS-CRP, BASFI, and ASAS HI were automatically generated and recorded. A radiologist specialized in the musculoskeletal system read the radiographic images of the C-spine and L-spine to calculate mSASSS.

### Study subjects

From October 23, 2017 to October 22, 2018, 360 patients with AS who completed ASAS HI assessments on at least one visit were consecutively selected from the TCVGH-AS cohort. Patients were excluded if they did not fill out questionnaires required for the assessments of BASDAI, ASDAS or BASFI even though they completed ASAS HI assessments. The main reason of not completing the questionnaires was inadequate time. However, we did not record the number of patients who did not complete the questionnaires and the exact reasons for not completing the questionnaires. The last date of assessment was identified as the index date. Of the 360 patients, 53 patients did not have mSASSS data and were excluded. Finally, 307 AS patients were enrolled into the study as subjects.

## Outcome

The outcome was a total ASAS HI score.

## Potential confounders

To our knowledge, no prior studies have investigated risk factors for the sum score of ASAS HI. Given that ASAS HI was validated to assess health and function in patients with SpA, we only considered the variables that were both shown to be risk factors for function/disability [38–46], health status [42], or health-related quality of life [47, 48] in previous studies and available in our data as potential confounder. Therefore, potential confounders in the relationships between gender and ASAS HI include disease activity [38, 40, 45, 47–49], mSASSS [45, 49], age at the index date [42], disease (symptom) duration [38, 42, 43, 48], smoking history (never, ever/current) [39, 43, 46], a family history (first degree or secondary degree relatives) of AS [42, 43], number of comorbidities (including hypertension, diabetes mellitus, hyperlipidemia, hepatitis B, hepatitis C, gout, coronary artery disease, stroke, periodontal disease, osteoporosis) [43, 50], and the presence of any extra-spinal manifestations (i.e., uveitis, psoriasis, inflammatory bowel disease, peripheral arthritis, enthesitis, and dactylitis) [40, 50].

Given the existence of collinearity, and the moderate to high correlations among ESR, CRP, ASDAS-ESR, ASDAS-CRP, and BASDAI (Table A in S1 Table), we used ASDAS-ESR, ASDAS-CRP, BASDAI+ESR or BASDAI+CRP to represent disease activity.

**Potential mediators.** Given that disease activity and structual damage of the spine were the most important risk factors for poor health and function [45, 47] and were correlated with gender (Table A in S1 Table), we also examined whether or not disease activity measures (i.e., ESR, CRP, ASDAS-ESR, ASDAS-CRP and BASDAI) and mSASSS had potential mediation effects for the influence of gender on ASAS HI.

## Statistical analysis

Continuous variables were reported as a mean ± standard deviation (SD) and categorical variables were reported as a percentage of patients. Differences in continuous variables were examined using the Student's $t$-test, while categorical variables were examined using the Pearson's $\chi^2$ test. After adjusting for potential confounders, the associations between gender and ASAS HI, ASDAS-CRP, BASFI, and mSASSS, respectively were quantified by estimating the coefficients $B$ with 95% confidence intervals (CIs) using linear regression analyses. The statistical significance of the associations between each covariate and the outcome was examined by univariable linear regression analyses. In the multiple linear regression analyses, covariates included gender, age, disease duration, smoking status, number of comorbidities, extra-spinal manifestations, family history of AS, mSASSS, disease activity (model A, ASDAS-ESR; model B, ASDAS-CRP; model C, BASDAI+ESR; model D, BASDAI+CRP). For all covariates with a p-value < 0.2, variance inflation fractions (VIF) were calculated to determine the degree of collinearity. Apart from gender, disease activity and mSASSS, we excluded all other covariates with a VIF ≥ 10. The Akaike information criterion (AIC) was calculated for each model based on different definition of disease activity. We selected the final models with the best model fit according to AIC (i.e., the least AIC) [51]. A two-tailed $p$-value <0.05 was considered to be statistically significant. All statistical analyses were performed using the SAS statistical software, version 9.3 (SAS Institute, Inc., Cary, NC, USA).

To examine the potential mediation effects of disease activity measures (i.e., ESR, CRP, ASDAS-ESR, ASDAS-CRP and BASDAI) and mSASSS for the influence of gender on ASAS HI, we examined the three following regression equations: (1) regressing each potential mediators on gender; (2) regressing ASAS HI on gender; (3) regressing ASAS HI on both gender and

the potential mediator which had a significant association with gender shown in first equation [52]. The mediation effect was significant if the following four conditions occurred concurrently: (1) gender must affect the potential mediator in the first equation; (2) gender must be shown to influence ASAS HI in the second equation; (3) the potential mediator must affect ASAS HI in the third equation; (4) If all above 3 conditions existed, then the effect of gender on ASAS HI must be less in the third equation than in the second [52].

**Sensitivity analysis.** We conducted sensitivity analyses by transforming outcome variables from continuous variables to categorical variables and by varying the definitions of disease activity. An ASAS HI score of $\leq 5$ was indicative of good health, whereas a case of $>5$ was indicative of moderate to poor health [35]. Because disease activity is the most important confounding factor, we also conducted sensitivity using various definitions of disease activity (i.e., model A, ASDAS-ESR; model B, ASDAS-CRP; model C, BASDAI+ESR; model D, BASDAI +CRP). After adjusting for potential confounders, the association between gender and ASAS HI was quantified by calculating the odds ratios (ORs) with 95% CIs using multiple linear regression analysis and multivariable logistic regression analysis. Covariates were only considered as significant determinants for ASAS HI if the associations were consistently significant across all models in both multiple linear regression analyses and multivariable logistic regression analyses.

## Results

A total of 307 AS patients were enrolled for analyses, of which 245 (79.8%) were male, and 62 (20.2%) were female. The mean age was 46.4 ± 13.3 years. The mean disease duration was 20.6 ± 12.1 years. Table 1 compares the demographic data and clinical characteristics between males and females. Compared to females, males had a younger age at onset of symptoms and a longer disease duration. Males also had a higher percentage of patients with a current or previous history of tobacco use as well as a periodontal disease than females. Table B in the S1 Table compares the laboratory data and use of medication between males and females. Males had higher levels of creatinine, higher levels of alanine aminotransferase (ALT) (, and higher levels of hemoglobin. Males also had a higher percentage of patients with ALT > 40 IU/ml than females.

Table 2 revealed disease activity related measures, ASDAS, BASDAI, mSASSS, and BASFI in AS patients. Visual analogue scales of back pain, patient assessments of global health and fatigue, levels of ESR and ASDAS-ESR were higher in females compared with males. On the contrary, mSASSS was higher in males than in females. Table 3 compared the components and total scores of the ASAS HI between males and females. The sum score of ASAS HI was higher in females than males. Compared to males, a higher percentage of females had difficulties running and standing for long periods of time, experienced exhaustion often, reported lack of motivation to perform activities requiring physical effort, lost interest in sex, had difficulty concentrating, could not to wash their hair and reported being unable to overcome difficulties.

Table C in the S1 Table compares demographic data, clinical characteristics, disease activity related measures and mSASSS between AS patients with ASAS HI $\leq 5$ and AS patients with ASAS HI $> 5$. As shown in Table 4, using multiple linear regression analyses, males had lower sum scores of ASAS HI than females across all models (model C with the lowest AIC: $B =$ -1.91, 95% CI, −2.82−−1.00, p <0.001). As shown in Table 5, after multivariable logistic regression, males also had a lower risk of achieving scores of ASAS HI > 5 than females (model C with the lowest AIC: OR, 0.15, 95% CI, 0.07–0.36, p <0.001). ASDAS-ESR, ASDAS-CRP, and BASDAI were consistently associated with higher ASAS HI scores (Tables 4 and 5). Table D in the S1 Table displays the associations between gender and components of the ASAS HI. After

**Table 1. Demographic data and clinical characteristics at presentation of patients with AS.**

| | Total | Female | Male | |
|---|---|---|---|---|
| | n = 307 | n = 62 | n = 245 | P-value |
| Age, year, mean ± SD | 46.4 ± 13.3 | 44.9 ± 14.5 | 46.8 ± 13.0 | 0.333 |
| Disease duration, year, mean ± SD | 20.6 ± 12.1 | 15.7 ± 11.6 | 21.8 ± 12.0 | <0.001 |
| Smoking | 122 (39.7) | 4 (6.5) | 118 (48.2) | <0.001 |
| HLA-B27, n (%) | 300 (97.7) | 60 (96.8) | 240 (98.0) | 0.58 |
| Prior use of biologics | 19 (6.2) | 4 (6.4) | 15 (6.1) | 1.00 |
| Current use of biologics | 108 (35.5) | 28 (45.2) | 81 (33.1) | 0.08 |
| Current use of NSAIDs | 269 (87.6) | 54 (87.1) | 215 (87.8) | 0.89 |
| **Comorbidities** | | | | |
| No. of comorbidities, mean ± SD | 1.1 ± 1.2 | 0.7 ± 1.0 | 1.2 ± 1.2 | 0.001 |
| Hypertension | 77 (25.1) | 10 (16.1) | 67 (27.3) | 0.069 |
| Diabetes mellitus | 29 (9.4) | 3 (4.8) | 26 (10.6) | 0.165 |
| Hyperlipidemia | 55 (17.9) | 6 (9.7) | 49 (20) | 0.058 |
| Hepatitis B | 37 (12.1) | 6 (9.7) | 31 (12.7) | 0.520 |
| Hepatitis C | 8 (2.6) | 2 (3.2) | 6 (2.4) | 0.665* |
| Gout | 18 (5.9) | 2 (3.2) | 16 (6.5) | 0.544* |
| Coronary artery disease | 13 (4.2) | 1 (1.6) | 12 (4.9) | 0.478* |
| Stroke | 1 (0.4) | 0 (0) | 1 (0.4) | 1.000* |
| Periodontal disease | 77 (25.1) | 8 (12.9) | 69 (28.2) | 0.013 |
| Osteoporosis | 26 (8.5) | 6 (9.7) | 20 (8.2) | 0.702 |
| **Extra-spinal manifestation** | 165 (53.7) | 36 (58.1) | 129 (52.7) | 0.445 |
| Uveitis | 90 (29.3) | 18 (29.0) | 72 (29.4) | 0.956 |
| Psoriasis | 24 (7.8) | 4 (6.5) | 20 (8.2) | 0.795* |
| Crohn's disease | 0 (0.0) | 0 (0.0) | 0 (0.0) | - |
| Ulcerative colitis | 1 (0.3) | 1 (1.6) | 0 (0) | 0.202* |
| Peripheral arthritis | 87 (28.3) | 19 (30.6) | 68 (27.8) | 0.652 |
| Enthesitis | 57 (18.6) | 15 (24.2) | 42 (17.1) | 0.202 |
| Dactylitis | 9 (2.9) | 0 (0) | 9 (3.7) | 0.213* |
| Family history of AS (first or second degree relatives) | 125 (40.7) | 30 (48.4) | 95 (38.8) | 0.169 |
| First degree relatives | 59 (19.2) | 14 (22.6) | 45 (18.4) | 0.452 |
| Second degree relatives | 89 (29.0) | 22 (35.5) | 67 (27.3) | 0.207 |

Data were shown as number (percentage) unless specified otherwise.

*Fisher's exact test.

Abbreviations: AS, ankylosing spondylitis; SD, standard deviation.

adjusting for potential confounders, males had lower risks than females for the following: difficulty running and standing for long periods, frequent exhaustion, loss of interest in sex and difficulty washing their hair. As shown in Tables E–H in the S1 Table, multivariable analyses revealed that ASDAS-ESR, ASDAS-CRP, and BASDAI remained positively associated with ASAS HI scores in men and women with AS.

To test the potential mediation effects of disease activity measures and mSASSS for the influence of gender on ASAS HI, we firstly regress each potential mediator on gender (equation 1). As shown in Table I in S1 Table, gender was significantly associated with ESR, ASDAS-ESR and mSASSS. We then regressed ASAS HI on gender (equation 2) and ASAS HI on both gender and the potential mediator with a p-value < 0.05 in equation 1 (i.e., ESR, ASDAS-ESR and mSASSS) using linear regression analyses and logistic regression analyses. As

**Table 2. Disease activity related measures, ASDAS, BASDAI, mSASSS and BASFI in AS patients.**

| | Total | Female | Male | |
|---|---|---|---|---|
| | n = 307 | n = 62 | n = 245 | P-value |
| Back pain*# | 2.3 ± 1.9 | 2.7 ± 2.1 | 2.2 ± 1.8 | 0.040 |
| Peripheral joint pain*# | 1.5 ± 1.9 | 1.9 ± 2.1 | 1.3 ± 1.8 | 0.067 |
| Morning stiffness duration*# | 2.3 ± 2.7 | 2.0 ± 2.6 | 2.4 ± 2.7 | 0.344 |
| Patient's assessment of global health* | 1.6 ± 1.9 | 2.3 ± 2.1 | 1.4 ± 1.8 | 0.004 |
| Fatigue# | 2.8 ± 2.0 | 3.4 ± 2.2 | 2.7 ± 1.9 | 0.021 |
| Tenderness# | 1.3 ± 1.8 | 1.7 ± 2.2 | 1.2 ± 1.7 | 0.103 |
| Degree of morning stiffness# | 2.4 ± 2.2 | 2.2 ± 2.0 | 2.5 ± 2.3 | 0.361 |
| ESR | 12.5 ± 14.9 | 17.6 ± 12.9 | 11.1 ± 15.1 | 0.002 |
| CRP | 0.6 ± 1.6 | 0.4 ± 0.5 | 0.6 ± 1.7 | 0.228 |
| ASDAS-ESR | 1.6 ± 0.8 | 1.9 ± 0.8 | 1.5 ± 0.8 | <0.001 |
| ASDAS-ESR group, n (%) | | | | <0.001 |
| <1.3 | 131 (42.7) | 13 (21) | 118 (48.2) | |
| 1.3–2.1 | 108 (35.2) | 24 (38.7) | 84 (34.3) | |
| 2.1–3.5 | 56 (18.2) | 21 (33.9) | 35 (14.3) | |
| ≥3.5 | 12 (3.9) | 4 (6.5) | 8 (3.3) | |
| ASDAS-CRP | 1.5 ± 0.9 | 1.5 ± 0.9 | 1.5 ± 0.9 | 0.972 |
| ASDAS-CRP group, n (%) | | | | 0.414 |
| <1.3 | 139 (45.3) | 28 (45.2) | 111 (45.3) | |
| 1.3–2.1 | 104 (33.9) | 17 (27.4) | 87 (35.5) | |
| 2.1–3.5 | 50 (16.3) | 14 (22.6) | 36 (14.7) | |
| ≥3.5 | 14 (4.6) | 3 (4.8) | 11 (4.5) | |
| BASDAI | 2.0 ± 1.5 | 2.4 ± 1.8 | 2.0 ± 1.4 | 0.115 |
| mSASSS | 18.6 ± 22.2 | 6.0 ± 11.4 | 21.8 ± 23.1 | <0.001 |
| BASFI | 1.2 ± 1.7 | 1.1 ± 1.6 | 1.2 ± 1.7 | 0.765 |

Data were shown as mean ± standard deviation unless specified otherwise.

*ASDAS related measures.

#BASDAI related measures.

Abbreviations: ASDAS, ankylosing spondylitis disease activity score; BASDAI, Bath ankylosing spondylitis disease activity index; mSASSS, modified Stoke ankylosing spondylitis spinal score; BASFI, Bath ankylosing spondylitis functional index; ESR, erythrocyte sedimentation rate; CRP, C-reactive protein.

shown in Tables J and K in S1 Table, ESR had partial mediation effect given a less effect on ASAS HI in equation 3 than in equation 2, and ASDAS-ESR had complete mediation effect given a less effect on ASAS HI in equation 3 than in equation 2 and a non-significant effect of gender on ASAS HI when ASDAS-ESR was controlled in equation 3.

## Discussion

The aim of the study was to assess whether gender and other factors were associated with ASAS HI in AS patients using a single center, cross-sectional study design. To the best of our knowledge, this study is the first to investigate the association between gender and ASAS HI among AS patients. Using multiple linear regression analysis as well as multivariable logistic regression analysis, we found that female gender was an independent risk factor for higher total ASAS HI scores after adjusting for potential confounders. Regarding the associations between gender and components of the ASAS HI, as shown in Table 3, female patients tended to have worse scores mainly in the subjective features. Possible explanations included a higher prevalence of fibromyalgia and widespread pain in female patients with AS compared with

**Table 3. Sum score and components of ASAS HI among patients with ankylosing spondylitis.**

|  | Total | Female | Male | |
| --- | --- | --- | --- | --- |
|  | n = 307 | n = 62 | n = 245 | P-value |
| Sum score of ASAS-HI, mean ± SD | 4.7 ± 3.6 | 5.9 ± 3.8 | 4.3 ± 3.4 | 0.002 |
| Components of the ASAS HI |  |  |  |  |
| Pain sometimes disrupts my normal activities. | 197 (64.2) | 39 (62.9) | 158 (64.5) | 0.816 |
| I find it hard to stand for long. | 169 (55.0) | 43 (69.4) | 126 (51.4) | 0.011 |
| I have problems running. | 162 (52.8) | 40 (64.5) | 122 (49.8) | 0.038 |
| I have problems using toilet facilities. | 45 (14.7) | 12 (19.4) | 33 (13.5) | 0.242 |
| I am often exhausted. | 141 (45.9) | 39 (62.9) | 102 (41.6) | 0.003 |
| I am less motivated to do anything that requires physical effort | 124 (40.4) | 34 (54.8) | 90 (36.7) | 0.009 |
| I have lost interest in sex. | 34 (11.1) | 13 (21.0) | 21 (8.6) | <0.001 |
| I have difficulty operating the pedals in my car. | 11 (3.6) | 2 (3.2) | 9 (3.7) | 1.000* |
| I am finding it hard to make contact with people. | 28 (9.1) | 3 (4.8) | 25 (10.2) | 0.190 |
| I am not able to walk outdoors on flat ground. | 13 (4.2) | 4 (6.5) | 9 (3.7) | 0.332 |
| I find it hard to concentrate. | 62 (20.2) | 19 (30.6) | 43 (17.6) | 0.022 |
| I am restricted in traveling because of my mobility. | 93 (30.3) | 25 (40.3) | 68 (27.8) | 0.054 |
| I often get frustrated. | 45 (14.7) | 13 (21) | 32 (13.1) | 0.116 |
| I find it difficult to wash my hair. | 23 (7.5) | 11 (17.7) | 12 (4.9) | <0.001 |
| I have experienced financial changes because of my rheumatic disease. | 55 (17.9) | 11 (17.7) | 44 (18) | 0.968 |
| I sleep badly at night. | 167 (54.4) | 34 (54.8) | 133 (54.3) | 0.938 |
| I cannot overcome my difficulties. | 36 (11.7) | 13 (21) | 23 (9.4) | 0.011 |

Data were shown as number (percentage) unless specified otherwise.

*Fisher's exact test.

Abbreviations: ASAS HI, assessment of spondyloarthritis international society health index; SD, standard deviation.

male patients with AS, as prior studies reported [17, 18]. However, after adjusting for potential confounders, we found that females were at a higher risk than men for the following: difficulty running and standing for long periods of time, frequent exhaustion, loss of interest in sex, and difficulty washing their hair. Consistent with previous studies [21, 22, 24, 25, 49, 53, 54], we also found that females had higher scores for back pain, fatigue, and global health assessment. Also, consistent with most previous studies, this present study showed that female AS patients had lower mSASSS scores [27, 30, 55, 56], but similar ASDAS-CRP and BASFI scores compared with male AS patients [25, 54]. These gender-related differences may be explained by variations in biological factors, such as immune responses, genetics, sex hormones, and social or behavioral factors between male and female patients with AS [57]. The study also revealed that disease activity measures, including ASDAS-ESR, ASDAS-CRP, and BASDAI, were independent determinants for ASAS HI scores both in men and women with AS. Therefore, controlling disease activity might help improve the health status of AS patients regardless of gender.

We also found that ESR had partial mediation effect and ASDAS-ESR had complete mediation effect for the influence of gender on ASAS HI. A possible explanation was that compared with men, women had a lower hemoglobin level, which may lead to a higher ESR level as well as worse health status. However, other disease activity measures, including CRP, ASDAS-CRP, BASDAI, and mSASSS did not have mediation effects.

Taken together, the above findings suggest that female AS patients experienced worse overall functioning and health than male AS patients, when examined using the ASAS HI although female AS patients had less radiographic damage, with similar disease activities and physical function compared to male AS patients. The ASAS HI contains components that measure

**Table 4. Multiple linear regression analyses for determinants of ASAS HI sum scores in patients with ankylosing spondylitis.**

| Variable | Univariable | | | Multivariable[*] | | | | | | | |
| | B (95% CI) | P | VIF | Model A | | Model B | | Model C | | Model D | |
| | | | | B (95% CI) | P | B (95% CI) | P | B (95% CI) | P | B (95% CI) | P |
|---|---|---|---|---|---|---|---|---|---|---|---|
| Gender, male | −1.60 (−2.58– −0.62) | 0.002 | 1 | −1.48 (−2.47– −0.49) | 0.004 | −2.30 (−3.28– −1.13) | <0.001 | −1.91 (−2.82– −1.00) | <0.001 | −1.91 (−2.80– −1.02) | <0.001 |
| Age, year | 0.04 (0.01–0.07) | 0.005 | 2 | −0.02 (−0.06– 0.02) | 0.311 | 0.00 (−0.04– 0.04) | 0.894 | -0.01 (−0.05– 0.02) | 0.545 | -0.01 (−0.05– 0.03) | 0.546 |
| Disease duration, year | 0.05 (0.02–0.09) | 0.001 | 2 | 0.05 (0.01–0.09) | 0.014 | 0.04 (−0.001– 0.08) | 0.057 | 0.03 (-0.002– 0.07) | 0.066 | 0.03 (-0.002– 0.07) | 0.066 |
| Smoking, n (%) | 0.25 (−0.55– 1.07) | 0.541 | | 0.39 (-0.39– 1.17) | 0.322 | 0.30 (-0.49– 1.08) | 0.454 | 0.40 (-0.32– 1.11) | 0.273 | 0.40 (-0.32– 1.11) | 0.274 |
| No. of comorbidities | 0.43 (0.10–0.76) | 0.010 | 1 | 0.16 (-0.18– 0.51) | 0.357 | 0.16 (-0.18– 0.51) | 0.354 | 0.16 (-0.16– 0.48) | 0.325 | 0.16 (-0.16– 0.48) | 0.325 |
| Extra-spinal manifestation | 1.07 (0.27–1.86) | 0.009 | 1 | 0.16 (−0.56– 0.87) | 0.664 | 0.19 (−0.53– 0.91) | 0.610 | 0.14 (−0.52– 0.80) | 0.673 | 0.14 (−0.51– 0.80) | 0.673 |
| Family history of AS | 0.26 (-0.55– 1.08) | 0.529 | | 0.01 (-0.70– 0.73) | 0.975 | 0.04 (-0.68– 0.76) | 0.922 | -0.06 (-0.72– 0.60) | 0.864 | -0.06 (-0.72– 0.60) | 0.864 |
| ESR, mm/hour | 0.04 (0.01–0.07) | 0.003 | 5 | | | | | 0.00 (−0.02– 0.02) | 0.993 | | |
| CRP, mg/dl | 0.25 (−0.01– 0.50) | 0.057 | | | | | | | | 0.00 (−0.21– 0.21) | 0.999 |
| ASDAS-ESR | 2.14 (1.71–2.57) | <0.001 | 18 | 1.93 (1.48–2.37) | <0.001 | | | | | | |
| ASDAS-CRP | 1.77 (1.36–2.18) | <0.001 | 5 | | | 1.68 (1.28–2.08) | <0.001 | | | | |
| BASDAI | 1.38 (1.17–1.60) | <0.001 | 7 | | | | | 1.30 (1.08–1.52) | <0.001 | 1.30 (1.08–1.52) | <0.001 |
| mSASSS | 0.02 (0.01–0.04) | 0.007 | 1 | 0.02 (-0.002– 0.03) | 0.081 | 0.02 (-0.002– 0.03) | 0.090 | 0.02 (0.01–0.04) | 0.007 | 0.02 (0.01–0.04) | 0.007 |
| AIC | | | | 1003 | | 1007 | | 951[#] | | 951[#] | |

[*]Gender, age, disease duration, smoking, number of comorbidities, extra-spinal manifestations, family history of AS, mSASSS, and disease activity (model A, ASDAS-ESR; model B, ASDAS-CRP; model C, BASDAI+ESR; model C, BASDAI+CRP) were included in the multivariable analyses.

Abbreviations: ASAS HI, assessment of spondyloarthritis international society health index; P, p-value; CI, confidence interval; n, number; ESR, erythrocyte sedimentation rate; CRP, C-reactive protein; ASDAS, ankylosing spondylitis disease activity score; BASDAI, Bath ankylosing spondylitis disease activity index; mSASSS, modified Stoke ankylosing spondylitis spinal score, AIC, Akaike information criterion.

[#]The AIC in model C (951.35226) is trivially lower than that in model D (951.35234).

categories of pain, emotional function, sleep, sexual function, mobility, self-care, and community. Our findings suggested that physicians should pay more attention to the worse health status of female AS patients and try to improve ASAS HI by controlling disease activity with strategies such as adequate medical therapy, rehabilitation and psychosocial support.

The development of ASAS HI was aimed to quantify health not only in patients with AS but also in patients with non-radiographic axial SpA (nr-axSpA). However, previous studies showed that although both groups of patients did not differ regarding health status, they did differ in the proportions of males and signs of inflammation [58, 59]. Also, a certain proportion of patients with nr-axSpA did not progress to AS after many years of follow-up [59]. Taken together, the results of our study might not be applicable to patients with nr-axSpA.

Particular strengths of this study include comprehensive adjustment of potential confounders such as comorbidities, medications, and laboratory data, as well as the consistency of results obtained from sensitivity analysis. Regardless, some limitations of the study must be addressed. Firstly, using a single-center cohort have introduced selection bias, and provided a relatively small sample size for multiple covariates adjustment. However, the significant association between gender and the total ASAS HI score remained robust across various models of

**Table 5. Univariable and multivariable logistic regression analyses for determinants of ASAS HI > 5 in patients with ankylosing spondylitis.**

| | Univariable | | | Multivariable* | | | | | | | | |
|---|---|---|---|---|---|---|---|---|---|---|---|---|
| | | | | Model A | | Model B | | Model C | | Model D | | |
| Variable | OR (95% CI) | P | VIF | OR (95% CI) | P | OR (95% CI) | P | OR (95% CI) | P | OR (95% CI) | P | |
| Gender, male | 0.36 (0.20–0.64) | <0.001 | 1 | 0.25 (0.11–0.54) | 0.001 | 0.14 (0.06–0.32) | <0.001 | 0.15 (0.07–0.36) | <0.001 | 0.14 (0.06–0.33) | <0.001 | |
| Age, year | 1.03 (1.01–1.05) | 0.003 | 2 | 0.98 (0.95–1.01) | 0.263 | 1.00 (0.97–1.03) | 0.967 | 0.99 (0.95–1.03) | 0.453 | 0.99 (0.96–1.02) | 0.520 | |
| Disease duration, year | 1.03 (1.01–1.06) | 0.001 | 2 | 1.05 (1.01–1.09) | 0.009 | 1.04 (1.00–1.07) | 0.040 | 1.04 (1.00–1.08) | 0.047 | 1.03 (0.99–1.07) | 0.063 | |
| Smoking, n (%) | 1.01 (0.63–1.62) | 0.982 | | 1.12 (0.60–2.10) | 0.718 | 1.05 (0.57–1.95) | 0.877 | 1.18 (0.61–2.27) | 0.623 | 1.17 (0.61–2.24) | 0.648 | |
| No. of comorbidities | 1.34 (1.10–1.62) | 0.003 | 1 | 1.28 (0.98–1.67) | 0.071 | 1.27 (0.98–1.65) | 0.077 | 1.30 (0.98–1.72) | 0.066 | 1.30 (0.98–1.72) | 0.065 | |
| Extra-spinal manifestation | 1.70 (1.06–2.72) | 0.028 | 1 | 0.94 (0.53–1.69) | 0.845 | 0.96 (0.54–1.70) | 0.895 | 0.96 (0.52–1.76) | 0.891 | 0.99 (0.54–1.81) | 0.973 | |
| Family history of AS | 1.33 (0.83–2.14) | 0.230 | | 1.22 (0.69–2.18) | 0.491 | 1.23 (0.70–2.70) | 0.476 | 1.29 (0.70–2.36) | 0.410 | 1.29 (0.70–2.36) | 0.411 | |
| ESR, mm/hour | 1.03 (1.01–1.05) | 0.002 | 5 | | | | | 1.01 (0.99–1.04) | 0.255 | | | |
| CRP, mg/dl | 1.32 (0.98–1.77) | 0.065 | | | | | | | | 1.06 (0.81–1.39) | 0.664 | |
| ASDAS-ESR | 4.05 (2.73–6.02) | <0.001 | 18 | 4.15 (2.65–6.49) | <0.001 | | | | | | | |
| ASDAS-CRP | 2.72 (1.97–3.75) | <0.001 | 5 | | | 3.05 (2.11–4.39) | <0.001 | | | | | |
| BASDAI | 2.27 (1.83–2.83) | <0.001 | 7 | | | | | 2.47 (1.92–3.20) | <0.001 | 2.47 (1.91–3.19) | <0.001 | |
| mSASSS | 1.02 (1.01–1.03) | 0.004 | 1 | 1.02 (1.00–1.03) | 0.025 | 1.01 (1.00–1.03) | 0.031 | 1.02 (1.01–1.04) | 0.006 | 1.02 (1.01–1.04) | 0.005 | |
| AIC | | | | 325 | | 333 | | 306 | | 307 | | |

*Gender, age, disease duration, smoking, number of comorbidities, extra-spinal manifestations, family history of AS, mSASSS, and disease activity (model A, ASDAS-ESR; model B, ASDAS-CRP; model C, BASDAI+ESR; model C, BASDAI+CRP) were included in the multivariable analyses.

Abbreviations: ASAS HI, assessment of spondyloarthritis international society health index; OR, odds ratio; CI, confidence interval; P, p-value; VIF, variance inflation fraction; ESR, erythrocyte sedimentation rate; CRP, C-reactive protein; ASDAS, ankylosing spondylitis disease activity score; BASDAI, Bath ankylosing spondylitis disease activity index; mSASSS, modified Stoke ankylosing spondylitis spinal score; AIC, Akaike information criterion.

statistical analyses. Although the propensity score matching (PSM) is a method to reduce selection bis problems; however, it is important to include all confounding factors to balance the data for perfect conduction of PSM. If any critical covariates are neglected, the estimation would turn out to be severely biased results [60]. Consequently, this condition makes PSM impossible to be conducted in the present study because no prior studies had reported factors associated with ASAS HI. Secondly, the use of a crosss-sectional design may not have adequately reflected the longitudinal impact of disease activities. Third, some unmeasured potential confounding factors, such as diet, socioeconomic status, and concomitant fibromyalgia, were not included in the multivariable analyses. Finally, a single-center study limited the generalizability of the results.

## Conclusions

Using a single center, cross-sectional study, this is the first study to demonstrated that males had lower total ASAS HI scores than females in a Taiwanese AS cohort. Our findings suggest

that physicians should pay more attention to the worse health status in female AS patients and try to improve ASAS HI by controlling disease activity with strategies such as adequate medical treatment, rehabilitation and psychosocial support. Further longitudinal clinical studies are needed to confirm our findings and elucidate the underlying mechanisms.

## Supporting information

**S1 Table.**
(DOCX)

**S1 Dataset. Data of 307 AS patients.**
(SAV)

## Acknowledgments

The authors would like to thank the Biostatistics Task Force of Taichung Veterans General Hospital, Taichung, Taiwan, ROC for statistical support.

## Author Contributions

**Conceptualization:** Hsin-Hua Chen, Tsu-Yi Hsieh.

**Data curation:** Hsin-Hua Chen, Kuo-Lung Lai.

**Formal analysis:** Hsin-Hua Chen, Chin-Yin Huang.

**Funding acquisition:** Hsin-Hua Chen, Yi-Ming Chen, Yi-Hsing Chen.

**Investigation:** Yi-Ming Chen, Kuo-Lung Lai, Tsu-Yi Hsieh, Wei-Ting Hung, Ching-Tsai Lin, Chih-Wei Tseng, Kuo-Tung Tang, Yin-Yi Chou, Yi-Da Wu, Chia-Wei Hsieh, Wen-Nan Huang, Yi-Hsing Chen.

**Methodology:** Hsin-Hua Chen.

**Resources:** Yi-Ming Chen, Kuo-Lung Lai, Yi-Hsing Chen.

**Software:** Hsin-Hua Chen.

**Supervision:** Hsin-Hua Chen, Yi-Hsing Chen.

**Validation:** Hsin-Hua Chen, Yi-Ming Chen.

**Visualization:** Yi-Ming Chen.

**Writing – original draft:** Hsin-Hua Chen.

**Writing – review & editing:** Yi-Hsing Chen.

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
