## [Decision Letter · Decision Letter 0]

11 Feb 2020

PONE-D-19-34404

Gender Difference in ASAS HI among Patients with Ankylosing Spondylitis

PLOS ONE

Dear Dr Chen,

Thank you for submitting your manuscript to PLOS ONE. After careful consideration, we feel that it has merit but does not fully meet PLOS ONE’s publication criteria as it currently stands. Therefore, we invite you to submit a revised version of the manuscript that addresses the points raised during the review process.

Specifically, you should address all concerns raised by the reviewers, especially those raised by reviewer 2 where it is recommended a more concise manuscript with consideration of the objectives stated and results described. In addition the apparent conflicts of previous published work to be carefully addressed. 

 We would appreciate receiving your revised manuscript by Mar 26 2020 11:59PM. To enhance the reproducibility of your results, we recommend that if applicable you deposit your laboratory protocols in protocols.io, where a protocol can be assigned its own identifier (DOI) such that it can be cited independently in the future. For instructions see: http://journals.plos.org/plosone/s/submission-guidelines#loc-laboratory-protocols

We look forward to receiving your revised manuscript.

Kind regards,

Antony Nicodemus Antoniou, PhD

Academic Editor

PLOS ONE

Journal Requirements:

"This study did not receive any funding support."

"No"

3. Your ethics statement must appear in the Methods section of your manuscript. If your ethics statement is written in any section besides the Methods, please move it to the Methods section and delete it from any other section. Please also ensure that your ethics statement is included in your manuscript, as the ethics section of your online submission will not be published alongside your manuscript.

Reviewers' comments:

Reviewer's Responses to Questions

**Comments to the Author**

1. Is the manuscript technically sound, and do the data support the conclusions?

Reviewer #1: Partly

Reviewer #2: Partly

2. Has the statistical analysis been performed appropriately and rigorously? 

Reviewer #1: No

Reviewer #2: Yes

3. Have the authors made all data underlying the findings in their manuscript fully available?

Reviewer #1: Yes

Reviewer #2: No

4. Is the manuscript presented in an intelligible fashion and written in standard English?

Reviewer #1: No

Reviewer #2: No

5. Review Comments to the Author

Reviewer #1: 1. Concerns about data analysis

1) It has been previously reported that female AS patients tend to have more pain and low QoL despite of lesser radiographic progression of spine compared to male patients with AS.

And fibromyalgia is supposed to be one of the reason for that. But your data seem to be somewhat different from previous reports. Could you explain? Did you investigate the prevalence of fibromyalgia in this study?

2) mSASSS can be highly variable according to the evaluator. Who did mSASSS scoring? Was it performed by same evaluator? It needs to be described.

3) In table 1, does the prevalence of each manifestation mean ever present or at presentation?

4) In ASAS HI, female patients have worse score mainly in the subjective features. What do you think about it?

2. All participants were recruited from a single regional hospital, thus limiting the generalisability of the results. And there is discrepancy between the number of female and male patientsv enrolled. Thus, propensity score matching (PSM) would be better to control selection bias.

4. There are a few typos

- line 3, page 10 : antheses-> entheses

- line5 page 18,: thn-> than

Reviewer #2: In general: The manuscript describes a novel subject, the ASAS HI and its association with gender, and is interesting. The manuscript is too comprehensive, which makes the manuscript unclear. The authors investigate far more than their stated objective, namely to investigate the influence of gender on the ASAS HI. The manuscript would benefit from shortening the text, tables and variables.

Detailed feedback

Introduction:

1. The ASAS HI was developed to quantify health in patients with all forms of SpA, please adjust this in the text.

2. Please use more recent literature about gender and AS patients.

3. Do the authors think that having a longer delay for the AS diagnosis in female patients could explain worse outcomes in female patients? Please describe this shorty in the introduction.

Methods

4. Please describe if consecutive patients were included in the study or if a selection was made in the patients participating in the study. How many patients did not complete the questionnaires and were not included in the study and why? Please describe.

5. Why are patients excluded if they did not complete the BASDAI, ASDAS or BASFI questionnaires? The authors are investigating the influence of gender on the ASAS HI not the influence of gender on other questionnaires. This could introduce bias in the cohort.

6. Please match the objective to the methods or vice versa.

7. Please mention under data collection which EAMs, previous histories and commorbidities are investigated.

8. Too many variables have included in the analyses, which makes it very complex and a bit of a fishing expedition. Please consider less variables and/or make a correction for multiple testing. Please do not forget that the group of female patients is small and therefore not too many variables could be used in the analyses.

9. Please explain of each variable why they are considered to be confounder in the association between gender and ASAS HI, preferably using literature. A confounder should have a association with both the ASAS HI (outcome) and gender (dependent variable) and should not be in the causal path between gender and ASAS HI. If they not fulfill these criteria, a variable should not be considered to be a confounder. Or wanted the author to make a prediction model? This has different rules.

10. It is unclear to me why disease activity was investigated by four different measures and what the added value of this is.

Results

11. Please show the results of the association between gender alone and the ASAS HI. All analyses are corrected for disease activity, why could be in the causal path of the association.

12. The result could be shortened by removing the associations already mentioned in the tables.

13. Why are also the associations with BASFI and mSASSS investigated?

Table 3 is very interesting.

Discussion

13. What are the implication of the research on the clinical practice and describe how the worse health status could be improved in female AS patients.

14. Could the results also be applicable to axSpA patients (so radiographic and non-radiographic patients) and why?

6. PLOS authors have the option to publish the peer review history of their article (what does this mean?). If published, this will include your full peer review and any attached files.

Reviewer #1: No

Reviewer #2: No

---

## [Author Response · Author response to Decision Letter 0]

18 Mar 2020

Review’s Comments to the Author: 

Reviewer #1:

Concerns about data analysis

1. It has been previously reported that female AS patients tend to have more pain and low QoL despite of lesser radiographic progression of spine compared to male patients with AS.

And fibromyalgia is supposed to be one of the reason for that. But your data seem to be somewhat different from previous reports. Could you explain? Did you investigate the prevalence of fibromyalgia in this study?

Author response: 

(1) As shown in Table 2, female patients with AS had more pain than male patients with AS (2.7 ± 2.1 vs. 2.2 ± 1.8, p = .04). Regarding the ASAS HI sum scores, female patients also had worse health status (higher ASAS HI scores signify worse health status, as mentioned in line 2 on page 3) than male patients (5.9 ± 3.8 vs. 4.3 ± 3.4, p < 0.01) (Table 3). Also, as shown in Table 4 and Table 5, the multiple linear regression and multivariable logistic regression analyses consistently revealed that female patients with AS had worse health status as assessed by ASAS HI. Regarding radiographic progression, as shown in Table E in S1 Table, male patients with AS had a significantly higher mSASSS score than female patients with AS. Taken together, our data showed that female patients had more pain and lower health status (though not equal to QoL) despite less radiographic progression in the spine. This finding is compatible with previous reports.

(2) We did not investigate the prevalence of fibromyalgia in this study. We included this point in the last statement of study limitations in the Discussion section: “Third, some unmeasured potential confounding factors, such as diet, socioeconomic status, and concomitant fibromyalgia, were not included in the multivariable analyses.”

2. mSASSS can be highly variable according to the evaluator. Who did mSASSS scoring? Was it performed by same evaluator? It needs to be described.

Author Response: 

(1) All mSASSS scoring was assessed by the same radiologist specialized in the musculoskeletal system, which we had mentioned in the last sentence of the Data collection subsection in the Methods section: “A radiologist specialized in the musculoskeletal system read the radiographic images of the C-spine and L-spine to calculate mSASSS.”

3.In table 1, does the prevalence of each manifestation mean ever present or at presentation?

Author Response: 

(1) The prevalence of each manifestation in Table 1 means at presentation. 

(2) We revised our statement in the text on Data collection of the Methods section (lines 7-12 on page 5): “After the patients were registered, a trained nurse assisted rheumatologists in collecting information, including clinical characteristics, age at onset of symptoms, smoking status, previous histories, family histories, comorbidities, and clinical phenotype (i.e., the presence or absence of EAMs, peripheral arthritis, enthesitis, and dactylitis) at presentation.” 

(3) We also revised the title of Table 1 to read: “Demographic data and clinical characteristics at presentation of patients with AS.”

4.In ASAS HI, female patients have worse score mainly in the subjective features. What do you think about it?

(1) We have added related statements in the first paragraph (lines 13-21 on page 11 and lines 1-3 on page 12) of the Discussion section: “Regarding the association between gender and components of the ASAS HI, as shown in Table 3, female patients tended to have worse scores, mainly in the subjective features. Possible explanations included a higher prevalence of fibromyalgia and widespread pain in female patients with AS compared with male patients with AS, as prior studies reported (Rencber et al., 2019, Swinnen et al., 2018). However, after adjusting for potential confounders, we found that females were at a higher risk than men for the following: difficulty running and standing for long periods of time, frequent exhaustion, loss of interest in sex, and difficulty washing their hair. Also, consistent with most previous studies, this present study showed that female AS patients had lower mSASSS scores (Lee et al., 2007, van Tubergen et al., 2012, Ward et al., 2009, Ramiro et al., 2014), but similar ASDAS-CRP and BASFI scores compared with male AS patients (van der Horst-Bruinsma et al., 2013, Webers et al., 2016).These gender-related differences may be explained by variations in biological factors, such as immune responses, genetics, sex hormones, and social or behavioral factors between male and female patients with AS (Rusman et al., 2018).” 

5. All participants were recruited from a single regional hospital, thus limiting the generalisability of the results. And there is discrepancy between the number of female and male patients enrolled. Thus, propensity score matching (PSM) would be better to control selection bias.

Author Response:

(1) We cannot avoid selection bias in a single-center study simply by using propensity score matching (PSM). The main purpose of PSM is to control for confounding, especially confounding by an indication for two groups of patients receiving different treatment in observational data. It is crucial to include all confounding factors to conduct PSM perfectly. If any critical variables are omitted, the two comparison groups could be unbalanced, and the estimation would come up with severely biased results (Streiner and Norman, 2012). Consequently, this condition is impossible to employ in this study because no prior studies reported factors associated with ASAS HI. We added a statement regarding this limitation in the last paragraph of the Discussion section (lines 5 to 10 on page 13): “Although propensity score matching (PSM) is a method to reduce selection bias problems, it is important to include all confounding factors to balance the data to ensure that PSM is conducted perfectly. If any critical covariates are neglected, the estimation will result in being severely biased (Streiner and Norman, 2012). Consequently, this condition makes it impossible to conduct PSM in the present study because no prior studies reported factors associated with ASAS HI.”

(2) Regarding generalizability, we have added a statement for this limitation in the final sentence of the Discussion section: “Finally, a single-center study limited the generalizability of the results.”

6. There are a few typos

- line 3, page 10 : antheses-> entheses

- line5 page 18,: thn-> than

Author response: We have corrected these errors.

Reviewer #2: 

In general: The manuscript describes a novel subject, the ASAS HI and its association with gender, and is interesting. The manuscript is too comprehensive, which makes the manuscript unclear. The authors investigate far more than their stated objective, namely to investigate the influence of gender on the ASAS HI. The manuscript would benefit from shortening the text, tables and variables.

Detailed feedback

Introduction:

1. The ASAS HI was developed to quantify health in patients with all forms of SpA, please adjust this in the text.

Author response: 

(1) We have revised the statement in the Introduction section (lines 3-6 on page 5): “Recently, expert members of the Assessment of Spondyloarthritis International Society (ASAS) have developed a disease-specific questionnaire named the ASAS Health Index (ASAS HI) based on the Comprehensive International Classification of Functioning, Disability, and Health (ICF) Core Set, aimed to better quantify health in SpA (Kiltz et al., 2015).”

2. Please use more recent literature about gender and AS patients.

Author response: 

(1) We have used more recent literature about gender and AS in the Introduction section.

3. Do the authors think that having a longer delay for the AS diagnosis in female patients could explain worse outcomes in female patients? Please describe this shorty in the introduction.

Author response: 

(1) We describe this shortly in the Instruction section (from line 20 on page 4 to line 1 on page 5): “Prior studies revealed an association between the delay in AS diagnosis and worse function (Ibn Yacoub et al., 2012, Aggarwal and Malaviya, 2009), which might explain, at least in part, a worse function in female patients with AS compared with male patients with AS.”

Methods

4. Please describe if consecutive patients were included in the study or if a selection was made in the patients participating in the study. How many patients did not complete the questionnaires and were not included in the study and why? Please describe.

Author response: 

(1) We routinely assessed BASDAI, ASDAS-ESR, ASDAS-CRP, and BASFI in patients with AS from October 23, 2017, to October 22, 2018. A total of 360 patients completed the questionnaires. We analyzed the data of these patients retrospectively after their tracked personal information had been anonymized. The main reason for not completing the questionnaires was inadequate time. However, we did not record the number of patients who did not complete the questionnaires and the exact reason for not completing them. However, we provide a brief explanation regarding the reason for not completing the questionnaires in the subsection on “Study subjects” of the Methods section (from line 22 on page 6 to line 4 on page 7): “From October 23, 2017, to October 22, 2018, 360 patients with AS who completed ASAS HI assessments on at least one visit were consecutively selected from the TCVGH-AS cohort. The main reason for not completing the questionnaires was inadequate time. However, we did not record the number of patients who did not complete the questionnaires and the exact reasons for not completing the questionnaires.”

5. Why are patients excluded if they did not complete the BASDAI, ASDAS or BASFI questionnaires? The authors are investigating the influence of gender on the ASAS HI not the influence of gender on other questionnaires. This could introduce bias in the cohort 

Author response: 

(1) We assessed ASAS HI, BASDAI, ASDAS, and BASFI concurrently for each patient with AS, as mentioned in the subsection of “Data collection” in the Methods section. To avoid confusion, we revised the description in the subsection of “Study subjects”: “From October 23, 2017, to October 22, 2018, 360 patients with AS who received ASAS HI assessment on at least one visit were consecutively selected from the TCVGH-AS cohort.”

6. Please match the objective to the methods or vice versa.

Author response: 

(1) We revised the subsection of “Outcome” in the Methods section as: “The outcome was total ASAS HI score”. We have deleted the secondary outcomes, including BASFI and mSASSS, to match the methods to the objective.

7. Please mention under data collection which EAMs, previous histories and comorbidities are investigated.

Author response: 

(1) EAM included uveitis, psoriasis, and inflammatory bowel disease.

(2) Previous histories included previous tuberculosis history. However, to avoid confusion, we removed it, given that it is not a comorbidity and is not likely to be a confounder.

(3) Comorbidities included hypertension, diabetes mellitus, hyperlipidemia, hepatitis B, hepatitis C, gout, coronary artery disease, stroke, periodontal disease, and osteoporosis. We revised the statement in the subsection of “Data collection” in the Methods section: “After the patients were registered, a trained nurse assisted rheumatologists in collecting information including clinical characteristics, age at onset of symptoms, smoking status, previous tuberculosis history, family histories, comorbidities (hypertension, diabetes mellitus, hyperlipidemia, hepatitis B, hepatitis C, gout, coronary artery disease, stroke, periodontal disease, and osteoporosis) and extra-spinal manifestations (i.e., uveitis, psoriasis, and inflammatory bowel disease], peripheral arthritis, enthesitis, and dactylitis) at presentation.”

8. Too many variables have included in the analyses, which makes it very complex and a bit of a fishing expedition. Please consider less variables and/or make a correction for multiple testing. Please do not forget that the group of female patients is small and therefore not too many variables could be used in the analyses.

Author response: 

(1) We have reduced the number of variables in the multiple linear regression and multivariable logistic regression analyses. As you mentioned in Question 9, we only considered potential confounders for ASAS HI in the causal path between gender and ASAS HI as covariates. However, no prior studies have investigated risk factors directly for the sum score of ASAS HI. 

(2) Because ASAS HI was validated to assess health and function in patients with SpA, we only considered the variables that were shown to be both significant risk factors for function, health status, or health-related quality of life in previous literature, and available in our data as potential confounders. We revised our description regarding potential confounders in the first paragraph of the subsection of potential confounders in the Methods section (lines 12-22 on page 7): “To our knowledge, no prior studies have investigated risk factors for the sum score of ASAS HI. Given that ASAS HI was validated to assess health and function in patients with SpA, we only considered the variables that were both shown to be risk factors for function/disability (Falkenbach et al., 2003, Zhang et al., 2018, Song et al., 2017, Schiotis et al., 2012, Forejtova et al., 2008, Ward et al., 2005, Machado et al., 2010, Landewe et al., 2009, Chung et al., 2012), health status (Forejtova et al., 2008), or health-related quality of life (van Lunteren et al., 2018, Law et al., 2018) in previous studies and available in our data as potential confounder. Therefore, potential confounders in the relationships between gender and ASAS HI include disease activity (van Lunteren et al., 2018, Law et al., 2018, Song et al., 2017, Falkenbach et al., 2003, Glintborg et al., 2010, Landewe et al., 2009), mSASSS (Glintborg et al., 2010, Landewe et al., 2009), age at the index date (Forejtova et al., 2008), disease (symptom) duration (Law et al., 2018, Forejtova et al., 2008, Ward et al., 2005, Falkenbach et al., 2003), smoking history (never, ever/current) (Zhang et al., 2018, Chung et al., 2012, Ward et al., 2005), a family history of AS (Forejtova et al., 2008, Ward et al., 2005), number of comorbidities (including hypertension, diabetes mellitus, hyperlipidemia, hepatitis B, hepatitis C, gout, coronary artery disease, stroke, periodontal disease, osteoporosis) (Ward and Kuzis, 2001, Ward et al., 2005), and the presence of any extra-spinal manifestations (i.e., uveitis, psoriasis, inflammatory bowel disease, peripheral arthritis, enthesitis, and dactylitis) (Ward and Kuzis, 2001, Song et al., 2017).”

9. Please explain of each variable why they are considered to be confounder in the association between gender and ASAS HI, preferably using literature. A confounder should have an association with both the ASAS HI (outcome) and gender (dependent variable) and should not be in the causal path between gender and ASAS HI. If they not fulfill these criteria, a variable should not be considered to be a confounder. Or wanted the author to make a prediction model? This has different rules.

Author response: 

(1) Please refer to the response to question 8. 

10. It is unclear to me why disease activity was investigated by four different measures and what the added value of this is.

Author response: 

(1) Because disease activity is the most important confounding factor, we also conducted sensitivity using various definitions of disease activity (i.e., model A, ASDAS-ESR; model B, ASDAS-CRP; model C, BASDAI+ESR; model D, BASDAI+CRP).

(2) We, therefore, revised the description in the subsection of “Sensitivity analysis” in the Methods section: “We conducted sensitivity analyses by transforming outcome variables from continuous variables to categorical variables, and by varying the definitions of disease activity. An ASAS HI score of ≤5 was indicative of good health, whereas a case of >5 was indicative of moderate to poor health (Kiltz et al., 2018). Because disease activity is the most important confounding factor, we also conducted sensitivity using various definitions of disease activity (i.e., model A, ASDAS-ESR; model B, ASDAS-CRP; model C, BASDAI+ESR; model D, and BASDAI+CRP). After adjusting for potential confounders, the association between gender and ASAS HI was quantified by calculating the odds ratios (ORs) with 95% CIs using multiple linear regression analysis and multivariable logistic regression analysis. Covariates were only considered as significant determinants for ASAS HI if the associations were consistently significant across all models in both multiple linear regression analyses and multivariable logistic regression analyses.”

Results

11. Please show the results of the association between gender alone and the ASAS HI. All analyses are corrected for disease activity, why could be in the causal path of the association.

Author response:

(1) Because the aim of the study was to investigate the associations of ASAS HI with gender and other factors, we showed the results of the associations between gender and other covariates with the ASAS HI. However, to avoid confusion, we have deleted the results regarding factors associated with BASFI and mSASSS.

(2) To our knowledge, no prior studies have investigated risk factors for the sum score of ASAS HI. Given that ASAS HI was validated to assess health and function in patients with SpA, we only considered the variables that were both shown to be risk factors for function/disability (Falkenbach et al., 2003, Zhang et al., 2018, Song et al., 2017, Schiotis et al., 2012, Forejtova et al., 2008, Ward et al., 2005, Machado et al., 2010, Landewe et al., 2009, Chung et al., 2012), health status (Forejtova et al., 2008), or health-related quality of life (van Lunteren et al., 2018, Law et al., 2018) in previous studies and available in our data as potential confounders. 

(3) According to the results of previous studies, disease activity is the most important factor with causal relationship with poor function/disability (van Lunteren et al., 2018, Law et al., 2018, Song et al., 2017, Falkenbach et al., 2003, Glintborg et al., 2010, Landewe et al., 2009), mSASSS (Glintborg et al., 2010, Landewe et al., 2009). Smolen et al. revealed that higher disease activity may lead to worse function and more structural damage, which again may lead to worse function (Smolen et al., 2018).

12. The result could be shortened by removing the associations already mentioned in the tables.

Author response: We have shortened the result section by removing the associations already mentioned in the tables.

13. Why are also the associations with BASFI and mSASSS investigated?

Table 3 is very interesting.

Author response: 

(1) To match the aim of the study and avoid confusion, we have removed the statement regarding factors associated with BASFI and mSASSS in the Results and Discussion sections.

(2) Regarding Table 3, we have added related statements to the first paragraph (lines 13-21 on page 11 and lines 1-3 on page 12) of the Discussion section: “Regarding the associations between gender and components of the ASAS HI, as shown in Table 3, female patients tended to have worse score mainly in the subjective features. Possible explanations included a higher prevalence of fibromyalgia and widespread pain in female AS patients compared with male AS patients, as prior studies reported (Rencber et al., 2019, Swinnen et al., 2018). However, after adjusting for potential confounders, we found that females had higher risks than men in the following: difficulty running and standing for long periods of time, frequent exhaustion, loss of interest in sex, and difficulty washing their hair. These gender-related differences may be explained by variations in biological factors, such as immune responses, genetics, sex hormones, and social or behavioral factors between male and female patients with AS (Rusman et al., 2018).”

Discussion

13. What are the implication of the research on the clinical practice and describe how the worse health status could be improved in female AS patients.

Author response: 

(1) We added the results of multiple linear and multivariable logistic regression analysis for factors associated with ASAS HI stratified based on gender in Table E to H in the S1 Table. The results showed that disease activity was positively associated with ASAS HI in both males and females. Therefore, controlling disease activity by strategies such as adequate medical treatment, rehabilitation and psychosocial support might improve ASAS HI in female AS patients

(2) We have mentioned the implication of the research on the clinical practice and described how the worse health status could be improved in female AS patients (lines 11-13 on page 11): “Our findings suggested that physicians should pay more attention to the worse health status of female AS patients and try to improve ASAS HI by controlling disease activity with strategies such as adequate medical therapy, rehabilitation and psychosocial support. “

(3) We also described the implication of the research on clinical practice and describe how to improve health status in the Conclusions subsection in the Discussion: “Our findings suggest that physicians should pay more attention to the worse health status in female AS patients and try to improve ASAS HI by controlling disease activity with strategies such as adequate medical treatment, rehabilitation and psychosocial support.”

14. Could the results also be applicable to axSpA patients (so radiographic and non-radiographic patients) and why?

Author response:

We added a paragraph (lines 15-20 on page 12) in the Discussion section to discuss this point: “The development of ASAS HI was aimed to quantify health not only in patients with AS but also in patients with non-radiographic axial SpA (nr-axSpA). However, previous studies showed that although both groups of patients did not differ regarding health status, they differed in the proportions of males and signs of inflammation (Malaviya et al., 2015, Kiltz et al., 2012). Also, a certain proportion of patients with nr-axSpA did not progress to AS after many years of follow-up (Kiltz et al., 2012). Taken together, the results of our study might not be applicable to patients with nr-axSpA.”

---

## [Decision Letter · Decision Letter 1]

5 May 2020

PONE-D-19-34404R1

Gender Difference in ASAS HI among Patients with Ankylosing Spondylitis

PLOS ONE

Dear Dt Chen,

Thank you for submitting your manuscript to PLOS ONE. After careful consideration, we feel that it has merit but does not fully meet PLOS ONE’s publication criteria as it currently stands. Therefore, we invite you to submit a revised version of the manuscript that addresses the points raised during the review process.

Please address the concerns raised by reviewer 2. I would draw your attention to point 3, which should be addressed as fully as possible. All other issues suggested should also be addressed on resubmission. 

We would appreciate receiving your revised manuscript by Jun 19 2020 11:59PM. To enhance the reproducibility of your results, we recommend that if applicable you deposit your laboratory protocols in protocols.io, where a protocol can be assigned its own identifier (DOI) such that it can be cited independently in the future. For instructions see: http://journals.plos.org/plosone/s/submission-guidelines#loc-laboratory-protocols

We look forward to receiving your revised manuscript.

Kind regards,

Antony Nicodemus Antoniou, PhD

Academic Editor

PLOS ONE

Reviewers' comments:

Reviewer's Responses to Questions

**Comments to the Author**

1. If the authors have adequately addressed your comments raised in a previous round of review and you feel that this manuscript is now acceptable for publication, you may indicate that here to bypass the “Comments to the Author” section, enter your conflict of interest statement in the “Confidential to Editor” section, and submit your "Accept" recommendation.

Reviewer #1: All comments have been addressed

Reviewer #2: (No Response)

2. Is the manuscript technically sound, and do the data support the conclusions?

Reviewer #1: Yes

Reviewer #2: Partly

3. Has the statistical analysis been performed appropriately and rigorously? 

Reviewer #1: Yes

Reviewer #2: Yes

4. Have the authors made all data underlying the findings in their manuscript fully available?

Reviewer #1: (No Response)

Reviewer #2: Yes

5. Is the manuscript presented in an intelligible fashion and written in standard English?

Reviewer #1: Yes

Reviewer #2: Yes

6. Review Comments to the Author

Reviewer #1: Thank you for your effort to revise manuscript and address comments raised sincerely. The manuscript describes the association of ASAS HI with gender, and is interesting.

Reviewer #2: The authors have addressed most of my comments. I have several comments:

1. Previous question 5. Where patients excluded if they did not fill out the BASDAI, BASFI or ASDAS questionnaire even though they did fill out the ASAS HI? If yes, please mention this in the methods.

2. Why were patients excluded if the mSASSS was not assessed? The main analysis is about the influence of gender on the ASAS HI and not about the influence of the mSASSS on the ASAS HI. The mSASSS is used as a confounder.

3. Previous question 11. I am concerned that disease activity might in the causal path between gender and the ASAS HI. This means that disease activity is no longer a confounder. If you correct the analyses for disease activity, the association between gender and ASAS HI might be biased. I understand that the authors would like to investigate the association between ASAS HI with gender and feel that disease activity is an important factor in this association. However, by adding all other factors from the univariable analysis, the effect of disease activity on the association between ASAS HI and gender is blurred by the other factors from the univariable analysis such as smoking. Please provide the results of the analysis between ASAS HI and gender with disease activity and no other factors. This would help to determine if disease activity is in the causal path or not. If not, it also shows the importance and the size of the influence of disease activity.

4. Some important features are now missing from Table 1 as they were removed. Please do not remove HLA-B27 status from Table 1 as it is an important SpA feature and the number of patients on NSAIDs and the number of patients on biologicals to get an idea about the study population.

5. Are the first seven variable of Table 2 measured as part of the ASDAS or BASDAI?

7. PLOS authors have the option to publish the peer review history of their article (what does this mean?). If published, this will include your full peer review and any attached files.

Reviewer #1: No

Reviewer #2: No

---

## [Author Response · Author response to Decision Letter 1]

15 May 2020

Review’s Comments to the Author: 

Reviewer #2: The authors have addressed most of my comments. I have several comments:

1. Previous question 5. Where patients excluded if they did not fill out the BASDAI, BASFI or ASDAS questionnaire even though they did fill out the ASAS HI? If yes, please mention this in the methods.

Author response: Thanks for your comments. We have mentioned this issue in Line 2-4 of the subsection of Study subjects in the Methods section: ‘Patients were excluded if they did not fill out questionaires required for the assessments of BASDAI, ASDAS or BASFI even though they completed ASAS HI assessments.’

2. Why were patients excluded if the mSASSS was not assessed? The main analysis is about the influence of gender on the ASAS HI and not about the influence of the mSASSS on the ASAS HI. The mSASSS is used as a confounder.

Author response: Thanks for your comments. To maximize the validity of the study, we only included patients with available data of major confounders (risk factors for poor health status and presence of data based association between males and females) including mSASSS as mentioned in previous studies (ref 45, 49). Also, as shown in our data, the mSASSS score was quite different between males and females (21.8 ± 23.1 vs. 6.0 ± 11.4, p <0.001). Therefore, we excluded patients without data of mSASSS. We have mentioned this issue in Line 1-11 of the subsection of Potential confounders in the Methods section: ‘Given that ASAS HI was validated to assess health and function in patients with SpA, we only considered the variables that were both shown to be risk factors for function/disability (Chung, Machado, van der Heijde, D'Agostino, & Dougados, 2012; Falkenbach, Franke, & van der Linden, 2003; Forejtova et al., 2008; Landewe, Dougados, Mielants, van der Tempel, & van der Heijde, 2009; Machado et al., 2010; Schiotis et al., 2012; Song, Wang, & Chen, 2017; Ward, Weisman, Davis, & Reveille, 2005; Zhang et al., 2018), health status (Forejtova et al., 2008), or health-related quality of life (Law et al., 2018; van Lunteren et al., 2018) in previous studies and availabe in our data as potential confounder. Therefore, potential confounders in the relationships between gender and ASAS HI include disease activity (Falkenbach et al., 2003; Glintborg et al., 2010; Landewe et al., 2009; Law et al., 2018; Song et al., 2017; van Lunteren et al., 2018), mSASSS (Glintborg et al., 2010; Landewe et al., 2009), age at the index date (Forejtova et al., 2008), disease (symptom) duration (Falkenbach et al., 2003; Forejtova et al., 2008; Law et al., 2018; Ward et al., 2005), smoking history (never, ever/current) (Chung et al., 2012; Ward et al., 2005; Zhang et al., 2018), a family history (first degree or secondary degree relatives) of AS (Forejtova et al., 2008; Ward et al., 2005), number of comorbidities (including hypertension, diabetes mellitus, hyperlipidemia, hepatitis B, hepatitis C, gout, coronary artery disease, stroke, periodontal disease, osteoporosis) (Ward & Kuzis, 2001; Ward et al., 2005), and the presence of any extra-spinal manifestations (i.e., uveitis, psoriasis, inflammatory bowel disease, peripheral arthritis, enthesitis, and dactylitis) (Song et al., 2017; Ward & Kuzis, 2001).’

3. Previous question 11. I am concerned that disease activity might in the causal path between gender and the ASAS HI. This means that disease activity is no longer a confounder. If you correct the analyses for disease activity, the association between gender and ASAS HI might be biased. I understand that the authors would like to investigate the association between ASAS HI with gender and feel that disease activity is an important factor in this association. However, by adding all other factors from the univariable analysis, the effect of disease activity on the association between ASAS HI and gender is blurred by the other factors from the univariable analysis such as smoking. Please provide the results of the analysis between ASAS HI and gender with disease activity and no other factors. This would help to determine if disease activity is in the causal path or not. If not, it also shows the importance and the size of the influence of disease activity.

Author response: 

(1) Thanks for your great comment! We examined whether or not disease activity measures (i.e., ESR, CRP, ASDAS-ESR, ASDAS-CRP and BASDAI) as well as mSASSS had potential mediation effects for the influence of gender on ASAS HI. We examined the correlations among disease activity measure, mSASSS, gender and ASAS HI (Table A in S1 Table).

(2) Regarding this issue, we added a subsection of Potential mediators in the Methods section: ‘Given that disease activity and structual damage of the spine were the most important risk factors for poor health and function (Landewe et al., 2009; van Lunteren et al., 2018) and were correlated with gender (Table A in S1 Table), we also examined whether or not disease activity measures (i.e., ESR, CRP, ASDAS-ESR, ASDAS-CRP and BASDAI) and mSASSS had potential mediation effects for the influence of gender on ASAS HI.’

(3) We mentioned the statistical method of testing mediation effect in the subsection of Statisical analysis in the Methods section: ‘To examine the potential mediation effects of disease activity measures (i.e., ESR, CRP, ASDAS-ESR, ASDAS-CRP and BASDAI) and mSASSS for the influence of gender on ASAS HI, we examined the three following regression equations: (1) regressing each potential mediators on gender; (2) regressing ASAS HI on gender; (3) regressing ASAS HI on both gender and the potential mediator which had significant association with gender shown in first equation (Baron & Kenny, 1986). The mediation effect was significant if the following four conditions occurred concurrently: (1) gender must affect the potential mediator in the first equation; (2) gender must be shown to influence ASAS HI in the second equation; (3) the potential mediator must affect ASAS HI in the third equation; (4) If all above 3 conditions existed, then the effect of gender on ASAS HI must be less in the third equation than in the second (Baron & Kenny, 1986).

(4) We added 3 tables (Table I-J in S1 Tables) to show the results of statistical analyses for potenial mediation effects. We also added a statement regarding these results in Line 1-9 on page 11 in the last paragraph of the Results section: ‘To test the potential mediation effects of disease activity measures and mSASSS for the influence of gender on ASAS HI, we firstly regress each potential mediator on gender (equation 1). As shown in Table I in S1 Table, gender was significantly associated with ESR, ASDAS-ESR and mSASSS. We then regressed ASAS HI on gender (equation 2) and ASAS HI on both gender and the potential mediator with a p-value < 0.05 in equation 1 (i.e., ESR, ASDAS-ESR and mSASSS) using linear regression analyses and logistic regression analyses. As shown in Table J and Table K, ESR had partial mediation effect given a less effect on ASAS HI in equation 3 than in equation 2, and ASAS-ESR had complete mediation effect given a less effect on ASAS HI in equation 3 than in equation 2 and a non-significant effect of gender on ASAS HI when ASDAS-ESR was controlled in equation 3.’

(5) We added a statement of discussion regarding the issue of mediation effect in Line 12-16 on page 12 in the second paragraph of the Discussion section: ‘We also found that ESR had partial mediation effect and ASDAS-ESR had complete mediation effect for the influence of gender on ASAS HI. Possible explanation was that compared with men, women had a lower hemoglobin level, which may lead to a higher ESR level as well as a worse health status. However, other disease activity measures including CRP, ASDAS-CRP, BASDAI, and mSASSS did not have mediation effects.’

4. Some important features are now missing from Table 1 as they were removed. Please do not remove HLA-B27 status from Table 1 as it is an important SpA feature and the number of patients on NSAIDs and the number of patients on biologicals to get an idea about the study population.

Author response: Thanks for your comments. We have added the data of HLA-B27, the number of patients on NSAIDs and the number of patients on biologics in Table 1.

5. Are the first seven variable of Table 2 measured as part of the ASDAS or BASDAI?

Author response: Yes. We have marked ASDAS related measures with ‘*’ and BASDAI related measures with ‘#’ immediately after the variables of Table 2, and mentioned this information in the footnote of Table 2: ‘*ASDAS related measures. #BASDAI related measures.’

---

## [Decision Letter · Decision Letter 2]

22 Jun 2020

Gender Difference in ASAS HI among Patients with Ankylosing Spondylitis

PONE-D-19-34404R2

Dear Dr. Chen,

We’re pleased to inform you that your manuscript has been judged scientifically suitable for publication and will be formally accepted for publication once it meets all outstanding technical requirements.

Kind regards,

Antony Nicodemus Antoniou, PhD

Academic Editor

PLOS ONE

Additional Editor Comments (optional):

Reviewers' comments:

Reviewer's Responses to Questions

**Comments to the Author**

1. If the authors have adequately addressed your comments raised in a previous round of review and you feel that this manuscript is now acceptable for publication, you may indicate that here to bypass the “Comments to the Author” section, enter your conflict of interest statement in the “Confidential to Editor” section, and submit your "Accept" recommendation.

Reviewer #2: All comments have been addressed

2. Is the manuscript technically sound, and do the data support the conclusions?

Reviewer #2: Yes

3. Has the statistical analysis been performed appropriately and rigorously? 

Reviewer #2: Yes

4. Have the authors made all data underlying the findings in their manuscript fully available?

Reviewer #2: Yes

5. Is the manuscript presented in an intelligible fashion and written in standard English?

Reviewer #2: Yes

6. Review Comments to the Author

Reviewer #2: (No Response)

7. PLOS authors have the option to publish the peer review history of their article (what does this mean?). If published, this will include your full peer review and any attached files.

Reviewer #2: No

---

## [Editor Report · Acceptance letter]

26 Jun 2020

PONE-D-19-34404R2 

Gender Difference in ASAS HI among Patients with Ankylosing Spondylitis 

Dear Dr. Chen:

I'm pleased to inform you that your manuscript has been deemed suitable for publication in PLOS ONE. Congratulations! Your manuscript is now with our production department. 

Kind regards, 

on behalf of

Dr. Antony Nicodemus Antoniou 

Academic Editor

PLOS ONE